# Facile Microfluidic Fabrication of Biocompatible Hydrogel Microspheres in a Novel Microfluidic Device

**DOI:** 10.3390/molecules27134013

**Published:** 2022-06-22

**Authors:** Minjun Chen, Ruqaiya Aluunmani, Guido Bolognesi, Goran T. Vladisavljević

**Affiliations:** Department of Chemical Engineering, Loughborough University, Loughborough LE11 3TU, UK; m.chen2@lboro.ac.uk (M.C.); ruqaya.alnaamani001@moe.om (R.A.); g.bolognesi@lboro.ac.uk (G.B.)

**Keywords:** hydrogel particles, microgel, droplet microfluidics, biocompatible microspheres, photopolymerization, swelling behaviour, free-radical polymerisation, poly(ethylene glycol) diacrylate, UV irradiation, entrapped nanoparticles

## Abstract

Poly(ethylene glycol) diacrylate (PEGDA) microgels with tuneable size and porosity find applications as extracellular matrix mimics for tissue-engineering scaffolds, biosensors, and drug carriers. Monodispersed PEGDA microgels were produced by modular droplet microfluidics using the dispersed phase with 49–99 wt% PEGDA, 1 wt% Darocur 2959, and 0–50 wt% water, while the continuous phase was 3.5 wt% silicone-based surfactant dissolved in silicone oil. Pure PEGDA droplets were fully cured within 60 s at the UV light intensity of 75 mW/cm^2^. The droplets with higher water content required more time for curing. Due to oxygen inhibition, the polymerisation started in the droplet centre and advanced towards the edge, leading to a temporary solid core/liquid shell morphology, confirmed by tracking the Brownian motion of fluorescent latex nanoparticles within a droplet. A volumetric shrinkage during polymerisation was 1–4% for pure PEGDA droplets and 20–32% for the droplets containing 10–40 wt% water. The particle volume increased by 36–50% after swelling in deionised water. The surface smoothness and sphericity of the particles decreased with increasing water content in the dispersed phase. The porosity of swollen particles was controlled from 29.7% to 41.6% by changing the water content in the dispersed phase from 10 wt% to 40 wt%.

## 1. Introduction

Hydrogels are hydrated three-dimensional networks of crosslinked hydrophilic polymers [1]. Their resemblance to living tissues, biocompatibility, high porosity, and the ability to absorb or release large quantities of water and aqueous solutions in response to changes in external conditions, such as pH, temperature, and ionic strength, opens up many exciting opportunities for their applications in drug delivery, tissue engineering, hygiene products (diapers, napkins, hospital bed sheets, sanitary towels), contact lenses, and food products [2,3].

Poly(ethylene glycol) diacrylate (PEGDA) is a hydrophilic and photopolymerisable synthetic polymer whose poly(ethylene glycol) (PEG) chains are functionalised with acrylate groups at both ends [4]. PEGDA is a promising hydrogel material due to its low toxicity, good biocompatibility, ease of chemical modification, and a broad range of possible molecular weights [4,5]. PEGDA hydrogels can be used as excipients in drug delivery systems [6] and wound dressings [7], carriers of biosensors [8], adsorbents of organic pollutants [9], and extracellular matrix mimics [10,11]. Bioactive substances, such as DNA, proteins, and fluorescently labelled avidin, have been encapsulated into shape-specific PEGDA nano- and microparticles to produce next-generation functional materials for sensing and drug delivery [12]. For example, PEGDA beads with tuned porosity were successfully used for multiplexed detection of mRNA [13], while PEG-based scaffolds were used in liver tissue engineering and allowed liver cells to maintain their function for a long time [14].

Fabrication methods of PEGDA micromaterials can be categorized into non-microfluidic methods and microfluidic methods. Non-microfluidic fabrication methods, including precipitation polymerisation [15], micro-moulding [16,17,18], electrohydrodynamic atomization [19], and extrusion through a nozzle [20], are limited in terms of particle uniformity, structural tunability, or size control, and some of them do not allow continuous production. On the other hand, microfluidic methods, such as droplet microfluidics, continuous flow lithography, and stop-flow lithography, not only provide precise control over the particle size and shape, but also allow a complex internal morphology, anisotropy, and tuneable chemical heterogeneity of the particles [21].

In lithography-based methods, PEGDA beads are generated in a poly(dimethyl-siloxane) (PDMS) microfluidic device using microscope-based sequential UV pulses sent through a photomask [22]. In its simplest form, the process requires the use of only one phase. The generated particles are non-spherical, and their shape and size are defined by the mask design and the shape of the channel filled with a stationary or flowing oligomer. In droplet-based microfluidic techniques, spherical PEGDA beads are fabricated in a two-phase system by polymerising droplets formed in microfluidic channels. Template droplets are usually generated using 2D PDMS or quartz glass microfluidic channels with a T-junction [23,24,25] or Ψ-junction [26,27] or 3D assemblies of borosilicate glass capillaries, hereinafter referred to as glass capillary devices [28,29,30,31].

Glass capillary devices do not require a photolithography process to be fabricated and are chemically highly robust [32], unlike PDMS devices that suffer from some solvent compatibility issues [33,34]. The most common flow arrangement of glass capillary devices for single-emulsion production is counter-current flow focusing [35]. However, fabrication of glass capillary devices is time-consuming and laborious due to the manual alignment of capillaries. In addition, just like PDMS chips, traditional glass capillary devices are disposable and difficult to clean because capillaries are permanently sealed with epoxy glue. Hence, there is an urgent need for reusable, glue-free glass capillary devices [36,37,38,39].

In this study, monodisperse PEGDA beads with controllable size and porosity were generated in a novel, reusable microfluidic device composed of glass capillaries and Lego^®^-inspired polymeric holders fabricated by CNC milling. The same Lego^®^-inspired holders were recently used to generate multiple emulsions in a different experimental set-up [40]. This is the first study to produce PEGDA beads in a glass capillary device with Lego^®^-inspired holders.

The utilisation of glue-free holders simplifies the alignment of the capillaries compared to the traditional device. The device was assembled using a round inner capillary and round outer capillary to achieve a perfect axisymmetric geometry [38], as compared to the quasi-axisymmetric geometry in traditional devices consisting of a round inner capillary and a square outer capillary [28]. Microgel production was optimised in terms of emulsion formulation and photopolymerization conditions. The particle size was manipulated by changing the orifice size of the collection capillary and the flow rates of the two phases. Additionally, morphological changes during droplet polymerisation and particle washing were systematically investigated using PEGDA solutions with different water contents. The produced PEGDA hydrogel particles were characterized using attenuated total reflection Fourier transform infrared (FTIR) spectroscopy, scanning electron microscopy (SEM), and fluorescence microscopy.

## 2. Experimental Section

### 2.1. Materials

PEGDA (poly (ethylene glycol) diacrylate, Sigma-Aldrich, Gillingham, UK, *Mw* 700) and 2-hydroxy-4′-(2-hydroxyethoxy)-2-methyl-propiophenone (Darocur 2959, Sigma-Aldrich) were used as a UV-curable prepolymer and photo-initiator, respectively. Silicone oil (XIAMETER^TM^ PMX-200, 100 cSt, Dow Corning, Midland, MI, USA) and XIAMETER^TM^ RSN-0749 resin (a 50/50 mixture of trimethylsiloxy-silicate and cyclomethicone from Dow Corning) were used as the carrier oil and hydrophobic surfactant, respectively. Octadecyltrimethyoxysilane (Sigma-Aldrich) was used as a hydrophobic modifier for internal glass capillary surfaces. Aqueous suspension of Firefly fluorescent red polystyrene nanoparticles (NPs) with a nominal diameter of 0.2 μm (Thermo Scientific^TM^ Fluoro-Max, 542/612 nm, 1% solids, Fisher Scientific, Loughborough, UK) was used in Brownian movement tracking experiments to monitor liquid–solid transition during photopolymerization. Ultrapure water was supplied using a Millipore Milli-Q Plus 185 water purification system.

The compositions of the dispersed phase used for the preparation of PEGDA beads are shown in Table 1. In all experiments, the continuous phase was silicone oil containing 3.5 wt% XIAMETER^TM^. Prior to their use, both solutions were sonicated in a Fisher Scientific FB15046 bath for 50 min to uniformly disperse all ingredients and remove air bubbles. In the NP diffusion study, the dispersed phase was composed of about 99 wt% PEGDA, 1 wt% photo-initiator, and 4.9 × 10^−4^ wt% of fluorescent polystyrene NPs.

### 2.2. Fabrication and Assembly of the Microfluidic Device

Lego-inspired blocks were designed using SolidWorks software (Dassault Systèmes, Vélizy-Villacoublay, France) and manufactured by CNC milling acetal plastic using a HAAS Super Mini mill (Norwich, UK). A 15 mm long outer capillary (World Precision Instruments, Hitchin, UK, I.D./O.D. = 1.56/2.0 mm) was mounted between the two Lego holders and fixed in place using screws, O-rings, and fasteners. An inner capillary (World Precision Instruments, I.D./O.D. = 0.58/1.00 mm, length = 152 mm) was pulled using a Flaming/Brown micropipette puller (P-97, Sutter Instruments Co., Novato, CA, USA), polished to the desired orifice size using abrasive paper, and rendered hydrophobic by dip-coating the tip in octadecyltrimethoxysilane. The prepared capillary with a hydrophobic tip was inserted into the outer capillary through the hole in the Lego holder so that its tip was placed in the midpoint of the outer capillary and then fixed in place using a stainless-steel tube connector. Figure 1a shows two detached Lego holders before assembly, while the assembled device is shown in Figure 1b.

### 2.3. Droplet Generation and Polymerisation Procedure

The assembled device was placed on an inverted microscope (GXM, GT Vision, Sudbury, UK), Figure 1c. The inlet tubing was attached to the device using stainless-steel tube connectors. The dispersed and continuous phase were delivered using Harvard, Apparatus 11 Elite syringe pumps (Biochrom Ltd, Cambridge, UK) from SGE gas-tight syringes (10 mL, Sigma-Aldrich, Gillingham, UK) using fine-bore Portex polyethylene medical tubing (0.86/1.52 mm I.D./O.D., Smiths Medicals, Luton, UK). All tubing was covered by aluminium foil to prevent a premature polymerisation of the PEGDA. Droplet generation in the device was observed with a Basler camera (Aca720–520uc, Ahrensburg, Germany) at a resolution of 728 × 544 pixels. The images were captured using a Basler Video Recording software at 25 frames per second. The generated droplets were collected in a Petri dish pre-filled with the continuous phase and polymerised by placing the Petri dish under a UV lamp (UVHAND 250GS Dr. Hönle AG, Gilching, Germany), Figure 1d. The UV light intensity was measured by a radiometer (365 nm, VLX-3W). After polymerisation, the particles were washed 5–7 times with acetone and deionised water to remove the oil phase.

### 2.4. Experimental Procedure for Droplet/Particle Size Measurement

The average particle/droplet diameter was determined using ImageJ2 v2.3.0 software (Wayne Rasband, National Institutes of Health, Kensington, Maryland, USA) using two different measurement methods. In the first method, the “Oval” tool in ImageJ was used to manually find the edges of all droplets, and droplet diameters were then calculated by ImageJ from their projected areas. The second method was based on using an ImageJ Plugin called Hough Circle Transform (HCT). Briefly, the edge of all droplets was roughly found using the “Find Edges” tool. At this point, each droplet had two prominent contours—one was the true external contour of the droplet, and the other was a small shadow of the droplet caused by non-vertical light from the optical microscope. To distinguish the two contours, the appropriate threshold was adjusted to highlight the real edge and fade the shadow line from the background. After that, the HCT was run to automatically collect all pixel values of the droplets. Then, the results were calibrated in microns and used to calculate the droplet diameters.

### 2.5. Droplet and Particle Characterisation

The bright-field images of the droplets/particles were taken using a CCD camera (Retiga 6000, Q-Imaging, Surrey, BC, Canada) interfaced to a computer running Q-capture software. Images of partially polymerised PEGDA particles loaded with fluorescent polystyrene NPs were captured by a fluorescence microscope (Eclipse TE300, Nikon, Surrey, UK), fitted with a CMOS camera (MQ013MG-ON, XIMEA) interfaced with a computer with XIMEA CamTool software. A filter cube (Nikon G-2A) with 510–560 nm excitation bandwidth and 590 nm long-pass emission filter was used for image acquisition. Attenuated total reflection Fourier transform infrared (ATR-FITR) spectroscopy was carried out using a spectrometer (Nicolet iS50, Thermo Scientific) in the wavelength range of 400–4000 cm^−1^ using a monolithic diamond crystal. The external morphology of the particles was analysed by a field emission scanning electron microscope (JEOL, JSM 7800F, Tokyo, Japan) operating at an accelerating voltage of 5.0 kV.

## 3. Results and Discussion

### 3.1. Droplet Generation Stability and Production Rate

A stable generation of monodispersed PEGDA-99 droplets in silicone oil is shown in Figure 1e. Silicone oil has several advantages over hydrocarbon oils, including excellent chemical and thermal stability [41] and higher biocompatibility [42]. Without a surfactant, droplets are prone to coalescence upon collision. Coalescence occurs when Waals forces between two approaching droplets exceed viscous and interfacial forces [43]. A surfactant creates a Marangoni stress that tends to immobilise the interface and inhibit film drainage and coalescence. Hence, a silicone-based surfactant, Xiameter^TM^ RSN-0749, was added to the silicone oil in various concentrations, 2 wt%, 3 wt%, and 3.5 wt%, to stabilise the droplets. The minimum Xiameter^TM^ concentration needed to fully stabilise PEGDA-99 droplets was 3.5 wt%, as shown in Appendix A. At the lower surfactant concentrations, some droplets formed clusters, which led to fused particles upon polymerisation. If the combination of hydrocarbon oil and surfactant is used, droplets can be fully stabilised using 4 wt% Span 80 dissolved in hexadecane [24].

Figure 2 shows the changes in size and generation frequency of PEGDA-99 droplets during continuous operation of the device at the continuous phase flow rate of 0.6 mL/h, the dispersed phase flow rate of 0.12 mL/h, and the orifice size of 400 µm. Clearly, the applied emulsion formulation and process conditions led to a stable droplet generation over 200 min, with an average droplet diameter of 282 µm and the coefficient of variation (CV) of droplet sizes of 1.6%. The average droplet generation frequency $\bar{f}$ was 2.8 Hz, as calculated from the mass balance equation for the dispersed phase: $\bar{f}=6Q_{d}/\left({\bar{d}}_{d}^{3}\pi \right)$, where $Q_{d}$ is the dispersed phase flow rate and ${\bar{d}}_{d}$ is the average droplet diameter.

### 3.2. Optimisation of Polymerisation Time

Figure 3 shows the morphology of PEGDA-99 particles after exposure of the droplets to UV light at 75 mW/cm^2^ for 10, 20, 30, 40, 50, and 60 s. UV irradiation for 10 s and 20 s did not cause any change in the droplet morphology, indicating that the curing had not started after 20 s. Xue at al. [24] successfully polymerised PEGDA droplets containing 99% PEGDA after UV irradiation for just 1.8 s. However, in that study, the droplets were polymerised on-fly in a serpentine downstream channel rather than in a Petri dish, and the UV light intensity was 125.4 mW/cm^2^. Off-chip polymerisation requires a longer time due to a thicker layer of emulsion in a Petri dish than in a microfluidic channnel.

After 30 s, the droplets acquired a core–shell morphology due to curing of the core region while the shell region was still uncured. It can be explained by the scavenging of radicals in the shell by dissolved oxygen diffused from the oil phase. It is well-known that oxygen is more soluble in silicone oil than in water, which can lead to a higher concentration of oxygen near the interface than in the droplet core. When oxygen reacts with a carbon-based free radical (R^•^), it converts the growing chain to an oxygen-based free radical according to the reaction: R^•^ + O_2_ → R-O-O^•^. The oxygen-based radical is less reactive than the carbon-based radical formed through the reaction between the growing chain and the monomer: R^•^ + M → R-M^•^, and it slows down the polymerisation reaction. To prevent this inhibition, various methods have been used, such as vacuum degassing, nitrogen purging, adding oxygen scavengers to the oil phase, creating an inert gas environment [25], and using high photo-initiator concentrations, >5 wt%, to create high radical concentrations that can consume all oxygen in the system. In this work, none of these techniques was attempted, yet the droplets were fully cured after 60 s, as can be seen by the progressive shrinkage of the uncured shell region and its complete disappearance after 60 s.

A complete solidification of droplets after 60 s was further demonstrated in Figure 4 by tracking trajectories of fluorescent polystyrene NPs in different regions within a droplet. The tracer NPs exhibited Brownian and convective motion in the unpolymerised shell region of the droplet after 30 s of UV irradiation, as can be seen by their irregular trajectories in Figure 4b. However, after 60 s, the NPs were totally immobilised in the same shell region and appear as stationary bright spots on the fluorescence image in Figure 4c. Moreover, the NPs were immobile in the core region after 30 s due to the fast solidfication of this region in the absence of oxygen inhibition (Figure 4a).

### 3.3. Impact of Orifice Size and Flow Rate Ratio on Droplet/Particle Diameter

Applications such as drug delivery require beads with a well-controlled size to achieve predictable release behaviour and high performance. As shown in Figure 5, highly uniform and non-aggregated particles with an average size in the range of 107–239 µm were produced using various orifice diameters, from 150 to 400 µm.

The droplets were exposed to UV light for 60 s at 75 mW/cm^2^ and left in silicone oil after curing for imaging. The droplet-to-orifice size ratio was kept in the range of 0.58–0.71 to prevent any contact between the emerging droplets and the wall of the inner capillary and, hence, to avoid any wetting problems. No difference in the mean droplet size was observed using the “Oval” tool in ImageJ and a Hough Circle Transform, as shown in Appendix A. Therefore, a Hough Circle Transform plugin can be safely used to achieve fast and automated particle size measurements.

Figure 6 shows size distribution histograms of particles produced using various orifice sizes at the same flow rate ratio $Q_{1}/Q_{2}$ of 6. The coefficient of variation was used as an indicator of particle size uniformity: $CV=\sigma /\bar{D}$, where σ is the standard deviation of particle sizes and $\bar{D}$ is the number-average particle diameter. The particles with a *CV* of 2–3% were obtained in all cases, except for the orifice size of 200 μm. According to the National Institute of Standards and Technology (NIST), particles are monodispersed if at least 90% of the particles have a size within 5% of the average size, $\bar{D}$. If the particle sizes are normally distributed, which can be assumed here, 90% of the particles will have a size within 1.64 standard deviations away from the mean. It means that, for monodispersed particles, 1.64σ ≤ 0.05$\bar{D}$, or CV ≤ 3%.

The effect of the flow rate ratio and orifice size on the size of PEGDA-99 droplets and particles is shown in Figure 7. The droplets were formed in the dripping regime and cured in 60 s. As can be seen, the droplet size can be controlled by adjusting the orifice size, and fine tuning can be achieved by changing the flow rate ratio. In Figure 7a, smaller droplets were generated at higher $Q_{1}/Q_{2}$ values, and the same trend can be seen in Appendix A. Droplet formation in the dripping regime is a result of the competition between the shear forces at the interface acting to pinch a droplet off the dispersed phase jet and the interfacial tension acting to prevent the pinch-off [44,45]. At $D_{o}$ = const, higher $Q_{1}/Q_{2}$ values lead to higher shear forces at the interface, which can more easily overcome the interfacial tension, and the jet break occurs at a smaller droplet size. The same trend with smaller droplets of PEGDA solution generated at higher external shear was observed in a microfluidic T junction device [24] and a co-flow glass capillary device in the presence of electric shear [30]. At $Q_{1}/Q_{2}$ = const, smaller droplets were produced at smaller orifice sizes, Figure 7b, which is in agreement with previously reported data obtained with different emulsions [40,46]. The effect of both parameters on ${\bar{d}}_{d}$ can be summarised by the following scaling low equation: ${\bar{d}}_{d}/D_{o}\;\propto \;{\left(Q_{1}/Q_{2}\right)}^{-x}$, where x is typically in the range of 0.37–0.40.

Another interesting aspect of the results presented in Figure 7 is that the droplets experienced a volumetric shrinkage between 1% and 4% during curing as a result of the sol–gel transition. In the liquid state, the prepolymer molecules are subjected to weak interactions due to van der Waals forces and hydrogen bonding that keep them apart at a distance of 0.3–0.4 nm [47]. However, during polymerisation, the acrylate double bonds of neighbouring molecules are converted to single covalent bonds with a length of ~0.15 nm [48], which means that PEGDA molecules are packed more tightly after curing.

### 3.4. Impact of Water Content in the Dispersed Phase on Shrinkage/Swelling Behaviour of Particles and UV Curing

In the work presented so far, the dispersed phase was a mixture of 99 wt% PEGDA and 1 wt% Darocur 2959, and the cured particles were kept in silicone oil for imaging and size measurements. In this section, the dispersed phase contained between 10 wt% and 40 wt% of water, and after curing, the particles were washed with acetone and transferred to deionised water. As shown in Figure 8, monodispersed droplets were obtained for all water contents investigated, and the particle size uniformity was not compromised after polymerisation and particle washing. The droplets became increasingly blurred when the water content in the dispersed phase increased from 10 wt% to 40 wt%, which can be explained by increasingly smaller differences in the refractive index of the dispersed phase and silicone oil. Indeed, the refractive index (RI) of pure XIAMETER™ PMX-200 Silicone Fluid at room temperature is 1.40 [49], while the RI of PEGDA 700 prepolymer is 1.49 [50]. Since water has an RI of ~1.33, the addition of water reduced the RI of the dispersed phase, and it became increasingly closer to that of the silicone oil. Indeed, the RI of PEGDA 700 with 10 wt% water is 1.47–1.48, and the RI of PEGDA 700 with 40 wt% water is 1.43–1.44 [50,51]. The RI of polymerised PEGDA is above 1.50 [50], so the cured particles are more distinctive on optical microscopy images than the droplets. Another noticeable feature in Figure 8 are volume changes during droplet curing and the subsequent transfer of the particles from silicone oil to water, which will be discussed in Figure 9.

Appendix A illustrates that PEGDA droplets with a higher water concentration required a longer exposure to UV light to be fully cured. The prepolymer solution with higher water content has a higher oxygen solubility and a decreasing number of acrylate functional groups per unit volume, which decelerates the propagation of polyacrylate chain and crosslinking. In addition, the reduced solution viscosity at a lower PEGDA concentration leads to an increased oxygen transport rate in the droplets [52]. The same trend was observed earlier. For example, at 70 mW/cm^2^, a complete on-fly polymerisation of the droplets containing 40% PEGDA and 6% Irgacure 1173 was impossible without oxygen purging [25] although the droplet residence time was up to 5 s, while droplets containing 99% PEGDA and 1% photoinitiator were fully cured after 1.8 s [24].

Volume changes during droplet curing and particle transfer into deionised water are shown in Figure 9. After curing, the particle volume $V_{2}$ represented 80%, 75%, 70%, and 68% of the initial droplet volume $V_{1}$ for the dispersed phase with 10 wt%, 20 wt%, 30 wt%, and 40 wt% water, respectively, as compared to 97–99% for the dispersed phase without any water. Shrinkage is inversely proportional to the molecular weight of the monomer units [53], which was not changed here, but it is also inversely proportional to the number of crosslinkable groups per unit volume, which is smaller at a higher water content in the dispersed phase. Therefore, in more dilute solutions, the average distance between prepolymer molecules is greater, and higher-volume contraction occurs during curing. After the particles were transferred from silicone oil to water, they swelled and reached the equilibrium volume $V_{3}$ in several hours. The water intake by hydrogel particles can be explained by the hydrophilic nature of their crosslinked polymer chains and their mesh-like structure. In the steady state, $V_{3}/V_{1}$ was 1.2, 1.1, 1.0, and 0.93 for the dispersed phase with 10 wt%, 20 wt%, 30 wt%, and 40 wt% water, respectively. The volume of swollen particles was 7% smaller than the initial droplet volume when the water content in the dispersed phase was 40 wt%, since a significant amount of water was expelled from the droplets during curing. When the water content in the dispersed phase was 10 wt% and 20 wt%, the swollen particles were larger than the initial droplets.

The porosity of swollen particles can be estimated from the equation:(1)$$\varepsilon_{3}=1-\frac{V_{p}}{V_{3}}=1-\frac{m_{p}/\rho_{p}}{V_{3}}=1-\frac{m_{d}x_{p}/\rho_{p}}{V_{3}}=1-\frac{\rho_{d}V_{1}x_{p}/\rho_{p}}{V_{3}}=1-\frac{\rho_{d}x_{p}/\rho_{p}}{V_{3}/V_{1}}$$
where $V_{3}$ is the volume of a swollen particle, $V_{p}$ the volume of pure polymer in the particle, $m_{p}$ the mass of pure polymer in the particle, $\rho_{p}$ the density of pure polymer, $m_{d}$ the initial droplet mass, $x_{p}$ the mass fraction of polymer in the droplet, $\rho_{d}$ the droplet density, and $V_{1}$ the initial droplet volume.

The swelling ratio Q is the fractional increase in the mass of the microgel due to water absorption and can be estimated based on the porosity of the swollen particle and the mass of pure polymer in the particle:(2)$$Q=\frac{m_{3}-m_{p}}{m_{p}}=\frac{\varepsilon_{3}\rho_{w}}{\left(1-\varepsilon_{3}\right)\rho_{p}}$$
where $m_{3}$ is the mass of swollen particles, and $m_{p}$ is the mass of dried particles (pure polymer). The porosity of swollen particles estimated from Equation (1) was found to increase from nearly 30% to more than 40% when the water content in the dispersed phase increased from 10 wt% to 40 wt% (Table 2). The swelling ratio calculated from Equation (2) increased from 0.36 to 0.60 as the water content in the PEGDA solution before polymerisation increased from 10 wt% to 40 wt%. The density of pure PEGDA in these calculations was assumed as $\rho_{p}$ = 1.18 g/cm^3^ [54]. The increase in swelling ratio with an increase in water content in a PEGDA prepolymer solution was observed in other studies. For example, the swelling ratio of PEGDA particles polymerised under green light in the presence of Eosin Y increased from 1.7 to 4.2 when the water concentration in the initial PEGDA solution increased from 60.5 vol% to 83.5 vol%. [55]. In this study, the swelling ratio was much smaller, which can be explained by a smaller concentration of water in the PEGDA prepolymer solution.

### 3.5. Real-Time ATR-FTIR and Morphology/Surface Topography Characterisation

For real-time analysis of the polymerisation kinetics of pure PEGDA 700, an in situ ATR-FTIR setup was used to record FTIR spectra at 3.8-s time intervals, as shown in Figure 10a,b. The mixture composed of 99 wt% of PEGDA 700 and 1 wt% of Darocur 2959 was placed in a closed chamber and exposed simultaneously to UV-A light and infrared radiation. The spectra in the 600–1800 cm^−1^ and 1600–1650 cm^−1^ wavenumber ranges exhibit the characteristic bands at 1064, 1186, 1636, and 1719 cm^−1^, corresponding to stretching aliphatic ether (C-O), ester (O=C-O), alkene double carbon (C=C), and α,β-unsaturated carbonyl (O=C-C=C) bonds of PEGDA.

The disappearance of alkene double carbon bond peaks at 1618 and 1636 cm^−1^ in Figure 10b occurred due to the conversion of the C=C to C-C bond initiated by benzoyl radicals from a UV-light-induced photo-initiator [56]. Similarly, the C=CH_2_ twisting peak at 1407 cm^−1^ almost completely disappeared after polymerisation. The peak absorbance at 1636 cm^−1^ was plotted against time in Figure 11c. The lamp was switched on after 10 s, and the polymerisation started after 12 s, i.e., just 2 s after the lamp was switched on. After 20 s, the absorbance already dropped from nearly 0.10 to 0.04 arbitrary units. After 50 s, the peak due to C=C bonds was negligible to the extent that it was covered by the residual background absorbance, which implies a near-to-100% conversion of double bonds.

Scanning electron microscopy (SEM) was carried out to investigate the surface morphology of PEGDA particles. Figure 11 shows SEM images of PEGDA particles produced using the dispersed phase containing between 0% and 50 wt% water. 

The images of particles produced using pure PEGDA (Figure 11(a_1_,a_2_)) and PEGDA with 30 wt% water (Figure 11(b_1_,b_2_)) reveal regular spherical shapes and extremely smooth surfaces with no cracks or dents. The mesh size of a PEG hydrogel made of 700 Da monomers is ~1.5 nm, which permits small molecules to penetrate the hydrogel network by diffusion [57], but prevents the release of cells, external vesicles, and NPs. However, the smoothness and sphericity of the particles decrease with increasing water content in the dispersed phase, as shown in Figure 11c,d. The reduced sphericity of the particles and higher surface roughness may be attributed to large volumetric shrinkage during curing, which can induce mechanical strain in the polymer [58] and surface deformations. The higher surface roughness could also be explained by the formation of a porous polymer matrix with sub-micron pore sizes [30,59].

The obtained results indicate that monodispersed PEGDA hydrogel beads with tuneable particle size, surface morphology, and swelling properties can be produced in a facile way using a simple glue-free glass capillary device that can easily be dismantled and reassembled, which offers many advantages compared to the traditional glass capillary device [28]. Indeed, the fabrication of conventional devices can be cumbersome, with device quality depending on the dexterity of the user who assembles them. Our novel microfluidic system solves this bottleneck by simplifying and deskilling the microcapillary device assembly. Furthermore, the generated droplets could be polymerized off-chip without oxygen purging in the device, which was required in some previous studies [25].

## 4. Conclusions

Monodispersed PEGDA particles in the size range from 100 µm to 250 µm were produced using a modular droplet microfluidic device, which is easy to assemble and disassemble and offers a facile continuous production of monodispersed PEGDA droplets over several hours. The droplet size in the microfluidic device was precisely controlled by adjusting the orifice size of the inner capillary, and fine tuning was achieved by changing the flow rate ratio of the continuous and dispersed phase. The PEGDA droplets experienced a significant volume contraction during curing by up to 33% for the PEGDA droplets containing 40 wt% water and then swelling by up to 50% when the cured particles were transferred from silicone oil into pure water. The swelling and shrinkage behaviour of PEGDA particles may have a significant impact on their morphology and the ability to release and absorb small molecules, such as nutrients, drugs, and metabolites. The droplet curing process was affected by the presence of dissolved oxygen and PEGDA concentration in the dispersed phase and was monitored in real time by in situ FTIR and tracking Brownian movements of fluorescent latex nanoparticles via fluorescence microscopy. The required UV exposure time and the particle sphericity, porosity, and surface roughness were significantly affected by the water content in the PEGDA solution before curing. The research highlights the importance of the optimisation of the emulsion formulation, the hydrodynamic conditions in a microfluidic device, and the photopolymerization conditions to produce PEGDA particles with desirable morphological characteristics. The developed method can be used to make hydrogel beads from any UV-curable hydrophilic polymer, including polyacrylamide, poly(ethylene glycol) methacrylate (PEGMA), and di(ethylene glycol) ethyl ether acrylate (DEGEEA). In future work, the produced beads will be loaded with TiO_2_, graphene and fluorescently labelled polymer latex nanoparticles to tailor their mechanical, catalytic, release, and imaging properties.

## Figures and Tables

**Figure 1 molecules-27-04013-f001:**
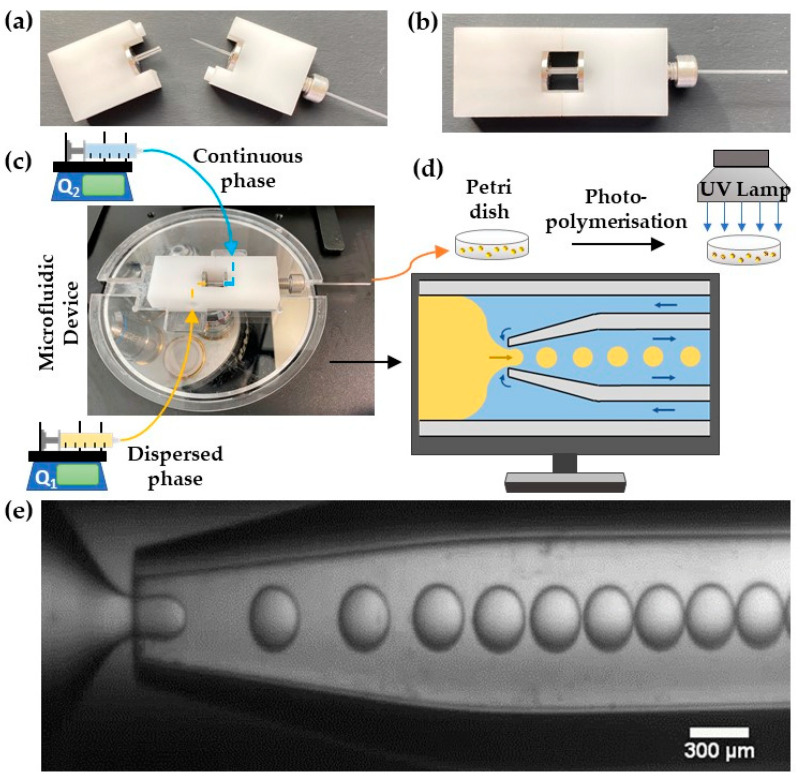
Photograph of (**a**) disassembled and (**b**) assembled Lego^®^-inspired glass capillary device. (**c**) The assembled device placed on a microscope stage. Q_1_ and Q_2_ are the flow rates of the dispersed and continuous phases, respectively. (**d**) A schematic view of droplet generation by counter-current flow focusing and droplet curing. (**e**) Video recording of droplet formation using 400 μm orifice at Q_1_ = 0.12 mL/h and Q_2_ = 0.6 mL/h.

**Figure 2 molecules-27-04013-f002:**
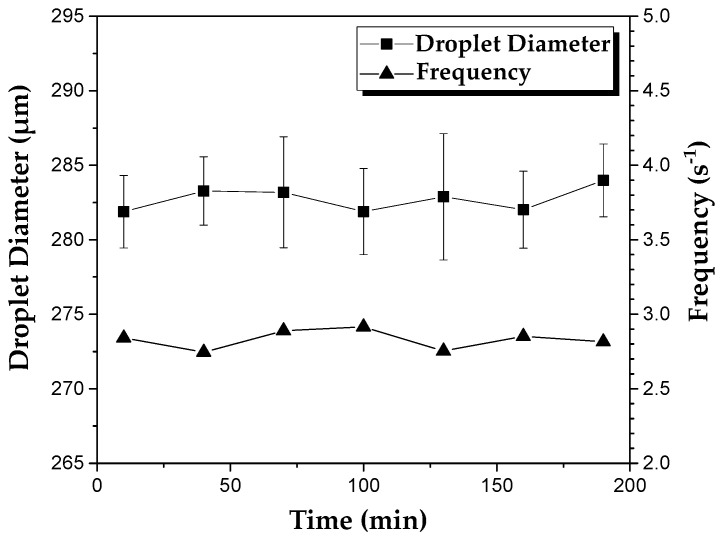
The variation of droplet diameter and the frequency of droplet generation over time in the device with an orifice size of 400 µm. The flow rates of the continuous and dispersed phase used were 0.6 mL h^−^^1^ and 0.12 mL h^−1^, respectively. The dispersed phase was composed of 99 wt% PEGDA and 1 wt% photo-initiator.

**Figure 3 molecules-27-04013-f003:**
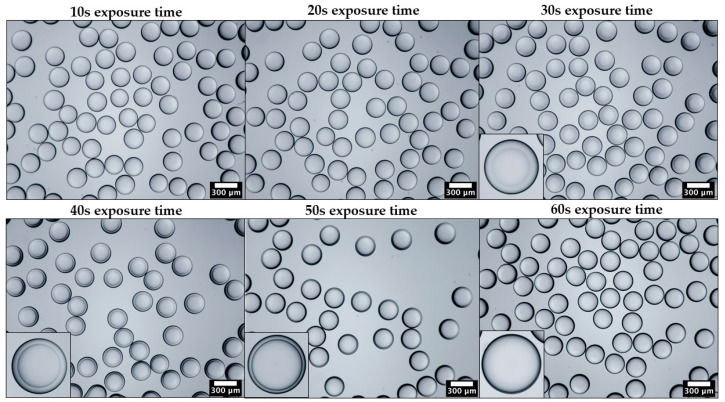
Optical microscopy images of PEGDA particles produced using different UV light exposure times and captured using 4× objective lens. The insets show magnified images of individual particles captured using 10× objective lens. The UV light intensity was 75 mW/cm^2^.

**Figure 4 molecules-27-04013-f004:**
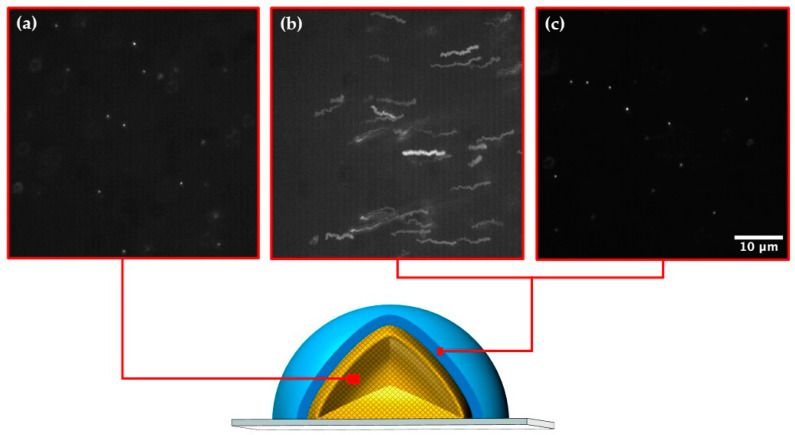
Schematic diagram of a partially photo-crosslinked PEGDA particle and epi-fluorescence micrographs of a “maximum intensity Z-projection” (processed on ImageJ) showing the trajectories of fluorescent polystyrene nanoparticles in different regions of the particle: (**a**) Polymerised core region after 30 s; (**b**) Unpolymerised shell region after 30 s; (**c**) Fully polymerized shell region after 60 s of UV light exposure at 75 mW/cm^2^.

**Figure 5 molecules-27-04013-f005:**
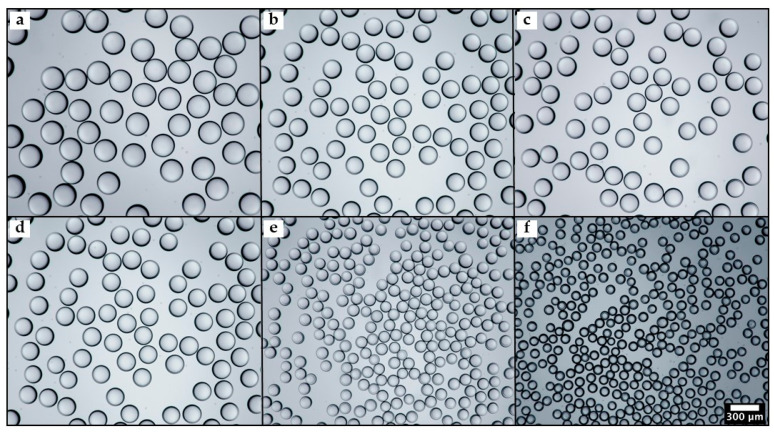
Optical microscopy images of PEGDA particles produced using inner capillary with different orifice diameters: (**a**) 400 µm; (**b**) 350 µm; (**c**) 300 µm; (**d**) 250 µm; (**e**) 200 µm; and (**f**) 150 µm. The average particle diameters are 239 µm, 194 µm, 191 µm, 145 µm, 122 µm, and 107 µm, respectively. The same scale bar applies to all images.

**Figure 6 molecules-27-04013-f006:**
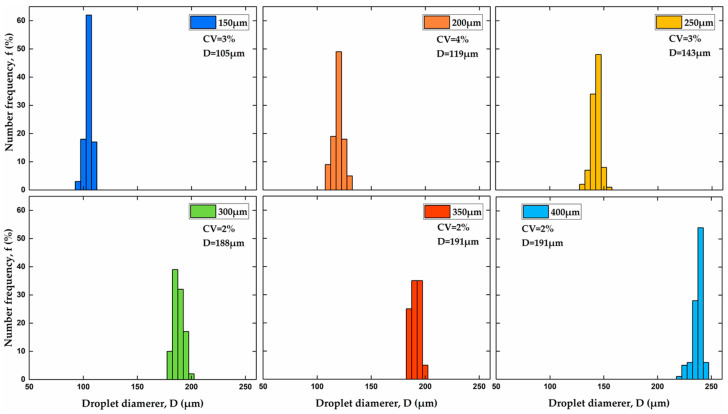
Size distribution of PEGDA particles produced using different orifice sizes shown in the legend of each graph. The coefficient of variation, CV and average particle diameter, $\bar{D}$ are shown in all graphs. The flow rate ratio, $Q_{1}/Q_{2}$ was kept constant at 6.

**Figure 7 molecules-27-04013-f007:**
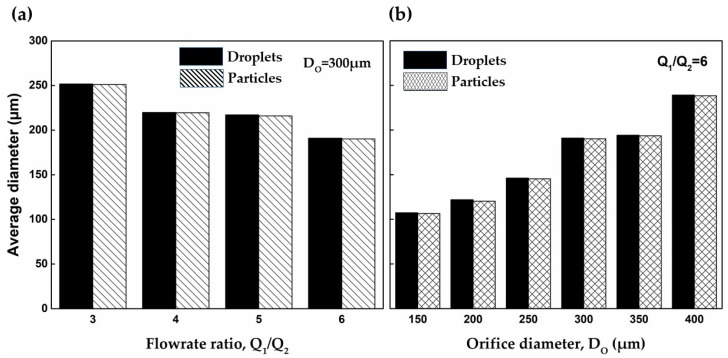
Comparison of the average diameter of droplets and particles: (**a**) The orifice size $D_{o}$ was kept constant at 300 μm. (**b**) The flow rate ratio $Q_{1}/Q_{2}$ was kept constant at 6 ($Q_{1}$ —continuous phase flow rate, $Q_{2}$ —dispersed phase flow rate). Error bars are too small to be visible.

**Figure 8 molecules-27-04013-f008:**
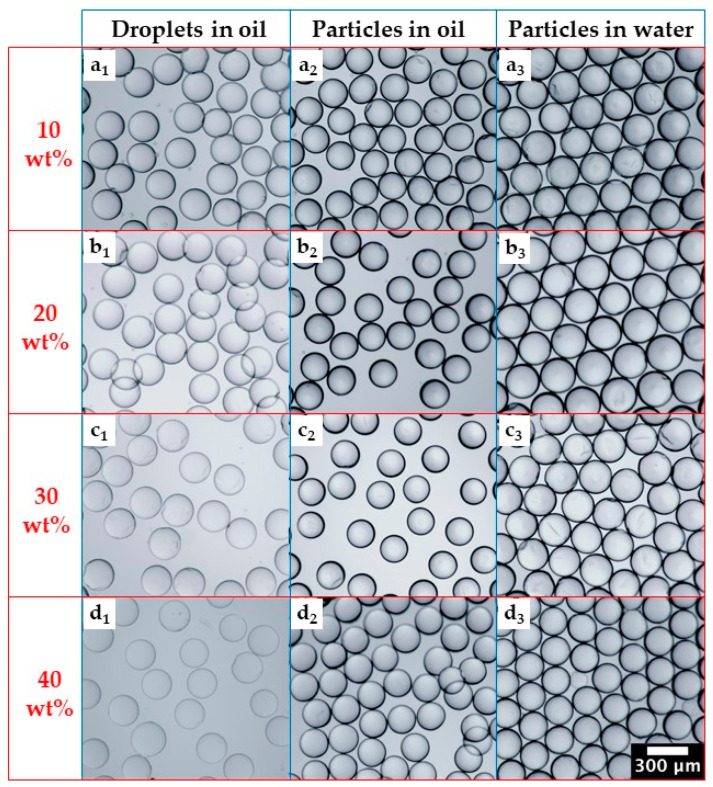
Optical microscopy images of droplets and PEGDA particles prepared using different water contents in the dispersed phase: (**a**) 10 wt%; (**b**) 20 wt%; (**c**) 30 wt%; (**d**) 40 wt%. Subscripts 1, 2, and 3 refer to collected emulsions before photopolymerization, PEGDA particles in silicone oil after photopolymerization, and PEGDA particles in DI water after washing with acetone. The same scale bar applies to all figures.

**Figure 9 molecules-27-04013-f009:**
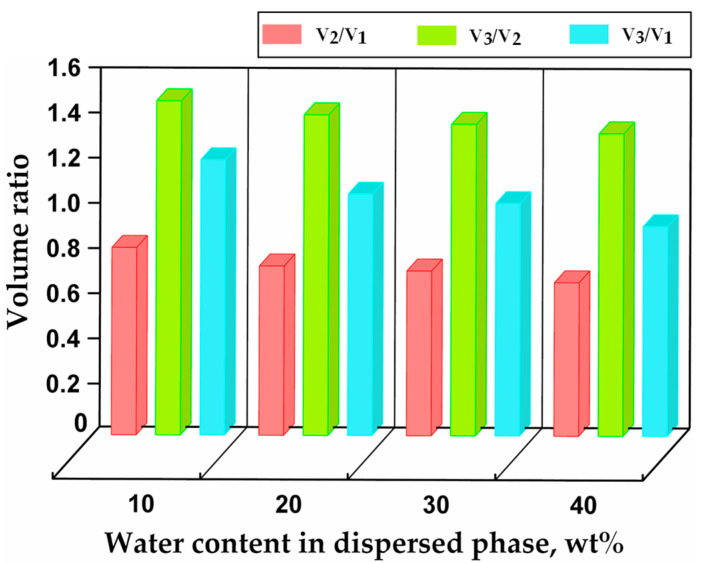
Relative changes in droplet/particle volume after polymerisation and swelling as a function of the initial water content in the dispersed phase. $V_{1}$ is the volume of droplets in silicone oil, *V*_2_ the volume of cured particles in silicone oil, and $V_{3}$ the volume of swollen particles in deionised water after swelling to the equilibrium state.

**Figure 10 molecules-27-04013-f010:**
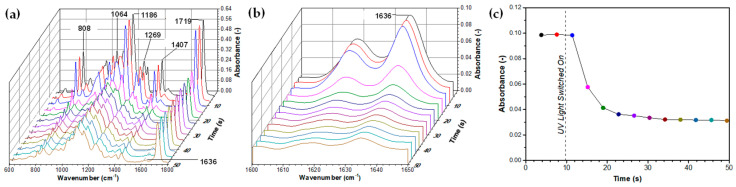
Real-time ATR-FTIR absorption spectra for UV-crosslinking of pure PEGDA in the presence of 1 wt% photo-initiator. The wavenumber regions are: (**a**) 600–1800 cm^−^^1^; (**b**) 1600–1650 cm^−1^. The disappearance of the peak at 1636 cm^−^^1^ with time due to the conversion of C=C bonds to C-C is shown in (**c**). The interval between spectra iterations was 3.8 s.

**Figure 11 molecules-27-04013-f011:**
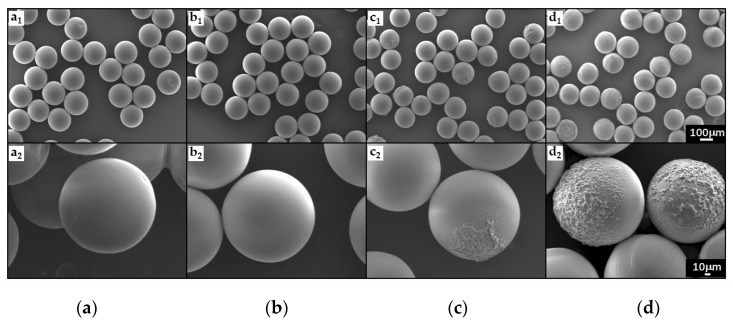
SEM images of PEGDA particles prepared with different amounts of water in the dispersed phase: (**a**) 0 wt%; (**b**) 30 wt%; (**c**) 40 wt%; (**d**) 50 wt%. 100 µm scale bar applies to figures (**a_1_**–**d_1_**), and 10 µm scale bar applies to figures (**a_2_**–**d_2_**).

**Table 1 molecules-27-04013-t001:** The dispersed phase compositions used for the fabrication of PEGDA microbeads.

Designation	Prepolymer (wt%)	Photo-Initiator (wt%)	Ultrapure Water (wt%)
PEGDA-99	99	1	0
PEGDA-89	89	1	10
PEGDA-79	79	1	20
PEGDA-69	69	1	30
PEGDA-59	59	1	40
PEGDA-49	49	1	50

**Table 2 molecules-27-04013-t002:** The porosity of swollen particles and swelling ratios for different water contents in the PEGDA prepolymer solution. The volume ratios of swollen particles to the initial droplets are also provided, as well as the droplet densities.

Water Content, wt%	Mass Fraction of Polymer, * *x_p_* (/)	Droplet Density, *ρ_d_* (g/m^3^)	*V*_3_/*V*_1_ (/)	Porosity after Swelling, *ε*_3_ (%)	Swelling Ratio, *Q* (/)
10	0.9	1.11	1.20	29.7	0.36
20	0.8	1.09	1.10	32.6	0.41
30	0.7	1.08	1.00	35.9	0.47
40	0.6	1.07	0.93	41.6	0.60

* The mass fraction of photo-initiator in the dispersed phase was neglected.

## Data Availability

The data presented in this study are available on request from the corresponding author.

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
