# Peer review of "Facile Microfluidic Fabrication of Biocompatible Hydrogel Microspheres in a Novel Microfluidic Device"

_molecules, 2022, doi:10.3390/molecules27134013_

Round 1
Reviewer 1 Report
This is a very nicely written manuscript that is quite informative. However, the method this manuscript describes seems to be known and published long time ago, e.g. DOI: 10.1002/anie.200604206
Also, PEGDA is a standard material for microgel formation.
Please comment on these points.
Author Response
We highly appreciate very useful comments of this reviewer. We have implemented their suggestions into the revised manuscript wherever possible.
Comments
Reviewer 1
This is a very nicely written manuscript that is quite informative. However, the method this manuscript describes seems to be known and published long time ago, e.g. DOI: 10.1002/anie.200604206
Response: The method described in DOI: 10.1002/anie.200604206 is based on using standard manually made glass capillary devices described by Utada et al. (2005) in DOI: 10.1126/science.1109164. Standard glass capillary devices are constructed by manually aligning two capillaries, the inner and outer one, and bonding the capillaries on the surface of a glass slide by epoxy glue. This fabrication process is time consuming, and the capillaries cannot be disassembled after use for cleaning, because they are permanently bonded on a glass plate. On the outer hand, our device is completely glue free and the capillaries can easily be dissembled since they are attached using LegoÒ-inspired plastic holders manufactured by CNC milling that can easily be separated and reassembled. Also, in our device the tubing for the delivery of inlet fluids is connected to the device via stainless-steel connectors, which are more convenient than syringe needles used to connect the tubing in manually made glass capillary devices. Finally, our method allows a perfect axisymmetric geometry for fluid flow because the device consists of a round inner capillary and a round outer capillary, while the standard device consists of a round inner capillary and a square outer capillary.
Also, PEGDA is a standard material for microgel formation.
Response: We fully agree with this opinion. However, in this study we have presented novel results and PEGDA beads are manufactured using a novel microfluidic device.
Reviewer 2 Report
This work reported the fabrication of the PEGDA microgels with tunable sizes based on a facile microfluidic fabrication. Lots of experiments have been done, most of which can support the topic of this work. This work is interesting but can be improved. I would like to recommend the publication of this work only if the authors address the following questions properly.
1) In the abstract, the author claims that the PEGDA microgels show tunable porosity, however, no relevant data can be found across the whole manuscript. Therefore, BET or corresponding data should be provided to demonstrate this.
2) The author proves that the PEGDA is used to fabricate microgels. Is it possible to fabricate microgels with other arylates? such as poly(ethylene glycol) methacrylate (PEGMA) and di(ethylene glycol) ethyl ether acrylate (DEGEEA). These works should be helpful and cited: Chemical Engineering Journal 429 (2022) 132342; ACS Appl. Mater. Interfaces 2020, 12, 45174-45183.
3) Fluorescent polystyrene NPs were used. The particle size of the polystyrene NPs and the molecular structures of dyes in polystyrene should be provided.
4) What are the possible applications of the PEGDA microgels?
Author Response
Reviewer 2
We highly appreciate very useful comments of this reviewer. We have implemented their suggestions into the revised manuscript wherever possible.
Comments
This work reported the fabrication of the PEGDA microgels with tunable sizes based on a facile microfluidic fabrication. Lots of experiments have been done, most of which can support the topic of this work. This work is interesting but can be improved. I would like to recommend the publication of this work only if the authors address the following questions properly.
- In the abstract, the author claims that the PEGDA microgels show tunable porosity, however, no relevant data can be found across the whole manuscript. Therefore, BET or corresponding data should be provided to demonstrate this.
Response: The porosity of swollen particles in our study was controlled from 29.7% to 41.6% by adjusting the water content in the initial PEGDA solution in the range from 10 wt% to 40 wt%, respectively. The porosity of swollen particles can be much higher than 42% using the dispersed phase with more than 40 wt% water or much smaller than 30% if the dispersed phase is pure PEGDA prepolymer. Therefore, the porosity of swollen particles is tuneable and can be precisely adjusted over a wide range to suit various applications.
In the revised manuscript, we have added the additional paragraph in Section 3.4 with a new table (Table 2) and a new equation (Equation 1) to discuss the porosity of swollen particles as a function of water content in the dispersed phase.
2) The author proves that the PEGDA is used to fabricate microgels. Is it possible to fabricate microgels with other arylates? such as poly(ethylene glycol) methacrylate (PEGMA) and di(ethylene glycol) ethyl ether acrylate (DEGEEA). These works should be helpful and cited: Chemical Engineering Journal 429 (2022) 132342; ACS Appl. Mater. Interfaces 2020, 12, 45174-45183.
Response: Our method is generic and can be used to make hydrogel particles composed of any UV curable hydrophilic polymer including PEGDA and DEGEEA. However, the emulsion formulation, operating conditions in the microfluidic device and photo-polymerisation conditions must be optimised for each monomer separately. We have added additional statement in the conclusion section to emphasise that the developed method can be used to generate hydrogel beads consisted of any UV curable hydrophilic monomer.
3) Fluorescent polystyrene NPs were used. The particle size of the polystyrene NPs and the molecular structures of dyes in polystyrene should be provided.
Response: The nominal particle diameter of fluorescent polystyrene NPs is 200 nm, and it is included in the revised paper. Unfortunately, the molecular structure of the dye was not revealed by the supplier. We believe the chemical structure of the dye does not affect the Brownian motion of NPs in the gel network.
4) What are the possible applications of the PEGDA microgels?
Response: In the future work, we will investigate diffusion and controlled release of various nanoparticles such as polymer latex nanoparticles, graphene, and liposomes from PEGDA beads. The release rate of nanoparticles depends on the mesh size of the gel network that will be controlled by the pH of the release medium. In addition, we will encapsulate TiO2 within the polymer matrix and use TiO2 embedded PEGDA beads as eco-friendly photocatalysts for degradation of organic molecules. Potential applications of nanoparticle-laden microgels, based on UV curable hydrophilic polymers, include – but are not limited to – tissue engineering, drug delivery, biosensing, environmental decontamination, hygiene and food products.
Round 2
Reviewer 1 Report
The corresponding author already published a similar, albeit more complex complex system (https://doi.org/10.1016/j.jcis.2021.12.094 ). Could you please state the principal differences between these devices? What is really new in the approach you present in the current manuscript?
Author Response
The corresponding author already published a similar, albeit more complex complex system (https://doi.org/10.1016/j.jcis.2021.12.094 ). Could you please state the principal differences between these devices? What is really new in the approach you present in the current manuscript?
Response: The device used in this study includes only one droplet break-up point and consists of two capillaries. Furthermore, this device is designed for production of single emulsions. The device used in doi: 10.1016/j.jcis.2021.12.094 includes two droplet break-up points and consists of three capillaries. The device is designed for the production of multiple emulsions.
Reviewer 2 Report
I am satisfied with the reversion and this work can be published now.
Author Response
Thank you very much for your decision.